# Structural optimization of drug molecules with incrementally trained language models

Tim Hörmann ⓘ , Domenic Mayer, Max Lewandowski ⓘ , Andrea Hunklinger ⓘ , Thomas Wein ⓘ & Daniel Merk ⓘ ✉

Automating structural optimization of drug molecules for on-target potency by machine learning is an open challenge in chemistry. Here, we capitalize on the ability of chemical language models (CLMs) to learn from sequential data and design new molecules with desired properties. We establish a training strategy mimicking the learning trajectory of a drug discovery program. Incremental CLM fine-tuning with increasingly potent template molecules from a given structure-activity relationship (SAR) series successfully biases the model to design highly active analogues. Prospective application of this technique to ligand development enables the data-driven design of molecules exceeding known representatives of given bioactive chemotypes in potency without external scoring. Our results reveal an ability of CLMs to capture SAR patterns and long-range dependencies, and to exploit SAR knowledge in designing analogues with improved on-target activity de novo corroborating their applicability to structural optimization of drug molecules.

Chemical language models (CLMs, Fig. 1)[1–5] process molecular information in string format and enable data-driven design of molecules with tailored features[6–8]. After pre-training on large corpora of molecules to capture the syntax of the chemical language Simplified Molecular Input Line Entry System (SMILES)[9] and general chemical properties[10,11], CLMs can capitalize on transfer learning[12] to extract molecular features relevant, e.g., for bioactivity on a biological target of interest, from a small set of fine-tuning molecules[13]. Thereby, these deep learning models allow navigation in the chemical space and are a very fruitful approach to computational de novo drug design[1,13–15]. For their ability to design new molecules with desired properties from scratch in a rule-free, data-driven fashion, CLMs have become an attractive tool to generate new chemical hypotheses and novel leads. De novo design with CLMs has yielded new molecules with bioactivity on intended macromolecular targets[11,16–18] highlighting their potential in drug discovery. However, achieving structural optimization, i.e., increasing the biological activity of a known ligand scaffold on a given target, with CLMs remains a challenge[15].

Optimizing a given template chemotype (i.e., molecular scaffold) for the intended modulation of a given biological target is a central objective in drug design and a challenging task. Wet-lab structural optimization typically requires extensive design-make-test cycles and

expert knowledge. Computational approaches performing well on this task are rare and generative molecular design, e.g., with CLMs, offers new opportunities to address this challenge. Recent approaches have improved analog design for given drug scaffolds in structural optimization tasks by generative models, e.g., by training CLMs with core structures and substituent groups[19,20] (i.e., scaffold decoration), by combining CLMs with matrix-based SAR formalism[15], by leveraging matched molecular pair data[21], and by fragment-based string modification[22,23]. Transformers have shown potential to optimize ADMET properties of drug molecules while being able to access a chemical space beyond matched pairs[24]. However, previous (generally theoretical) applications have mostly focused on rather trivial features such as logP and the quantitative estimate of drug likeness (QED) score which do not reflect the complexity of drug design[2,25,26]. The fewer attempts to solve the substantially more challenging task of optimizing a given template molecule for modulation of a given biological target with generative design have leveraged information on the target protein[27,28], used (potency-labeled) matched input pairs[26,29], or employed external scoring such as QSAR-based activity prediction[30,31] and docking[32,33]. Generative approaches to improve drug potency on a given biological target hence depend on (external) scoring of the computational designs using predictive models ("oracles"). However,

---

Department of Pharmacy, Ludwig-Maximilians-Universität München, Munich, Germany. ✉e-mail: daniel.merk@cup.lmu.de

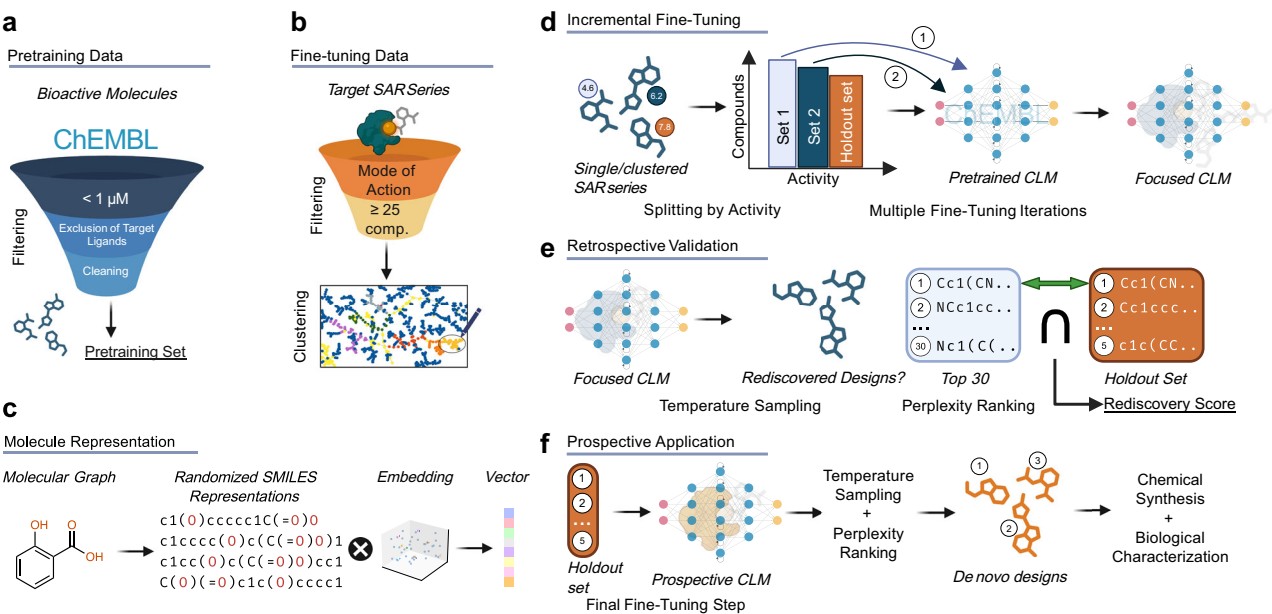

**Fig. 1 | Chemical language models (CLMs) and their application to structural optimization in this study. a** A target-ignorant CLM was pretrained with bioactive molecules sourced from ChEMBL[43] (bioactivity on any human target ≤1 μM) from which all molecules with an annotated activity on the target of interest as well as all structurally related molecules (Tanimoto similarity >0.4) were excluded to prevent any bias. **b** Fine-tuning of the target-ignorant baseline CLM was performed with series of known ligands for the target of interest sourced from BindingDB[74] and ChEMBL[43] which were clustered to identify entities of common structure-activity relationship (SAR) series. Each SAR series was limited to one mode-of-action (agonism/antagonism/inverse agonism on the target of interest) and to molecules with comparable data from uniform functional assays available to enhance intra-series comparability. Series with <25 molecules, high internal diversity (mean Tanimoto similarity <0.3), or unique molecules substantially differing from the rest of the series were excluded. **c** Molecules were represented as SMILES[9] strings which were processed with a word embedding layer converting individual characters into a multidimensional vector space. **d** To train the CLM for the chemical space of highly potent ligands of the target of interest, incremental fine-tuning was performed with subsets of an SAR series containing increasingly potent molecules. To observe the fine-tuning success of this approach in retrospective evaluation, the subset containing the most active molecules was excluded as holdout set. **e** After incremental fine-tuning with increasingly potent molecule sets, designs were generated from the CLM by multinomial sampling ($T=1.0$)[13] and ranked by perplexity[47], a model-intrinsic measure of design quality based on the model's uncertainty. For retrospective evaluation, the 30 top-ranking designs were compared with the holdout set, and a score was calculated based on the number of rediscovered highly active molecules. **f** Prospective application of incremental CLM training to structural optimization was performed utilizing full SAR series including the holdout molecules as final fine-tuning step. Top-ranking designs (by perplexity) were prepared and tested for modulation of the intended targets. Created in BioRender. Merk, D. (2026) https://BioRender.com/ewf6ktw.

comparative evaluation of recent generative algorithms found that more challenging tasks with limited oracle queries were not satisfactorily solved[34], and while many approaches may hold potential for prospective design, their experimental validation is mostly pending. Additionally, reliance on external oracles adds uncertainty as the quality of the predictive model for scoring is a critical performance factor[34], and high-quality predictive models (e.g., docking-based) may not be available for some targets of interest. Optimizing a given drug for a given biological target using CLMs without relying on external scoring has not been established but could open remarkable new opportunities for the application of generative design in drug discovery.

Here, we close this gap by mimicking design-make-test cycles of typical medicinal chemistry programs in the fine-tuning of CLMs thereby leveraging the potency ranking of known ligands for the target of interest as additional data dimension to create a learning trajectory. Using a long short-term memory (LSTM)[35,36] recurrent neural network, which has proven great performance for de novo design in various application-specific tasks and is the most popular architecture for CLMs[37], we capitalize on available structure-activity relationship (SAR) data and trained target-ignorant basic CLMs with increasingly potent entities of a given bioactive scaffold. We reveal an ability of CLMs to learn incrementally from the SAR data and benefit from the added dimension in the training data with improved performance in the rediscovery of highly active molecules. Using this approach for prospective applications enabled data-driven structural optimization

within series of known bioactive ligands (Fig. 1) of two pharmacologically relevant targets, the peroxisome proliferator-activated receptor (PPAR) γ and the retinoic acid receptor related orphan receptor (ROR) γ. These results corroborate incrementally fine-tuned CLMs for optimization of drug molecules without external scoring and extend the scope of CLMs in drug design.

## Results

### Incremental fine-tuning improves CLM performance in designing highly active molecules

The typical application of CLMs in de novo drug design encompasses a two-step training procedure to (i) capture the SMILES syntax in the pretraining and (ii) subsequently introduce a bias for the target of interest by fine-tuning[10,11]. We envisioned adapting this procedure to mimic the situation of a drug discovery effort starting from a hit with no SAR knowledge and learning from repeated design-make-test cycles. As proof-of-concept target we chose the ligand-activated transcription factor PPARγ for which many SAR datasets with chemically diverse scaffolds are available for broad exploration and application of our approach. PPARγ is a cellular lipid sensor with pivotal roles in adipose tissue, immune cells and the cardiovascular system as regulator of, e.g., metabolic balance, adipogenesis and macrophage function[38,39]. It is the target of antidiabetic thiazolidinediones (TZDs) and in the focus of drug development for various indications including metabolism-associated steatohepatitis and neurodegenerative diseases[40–42].

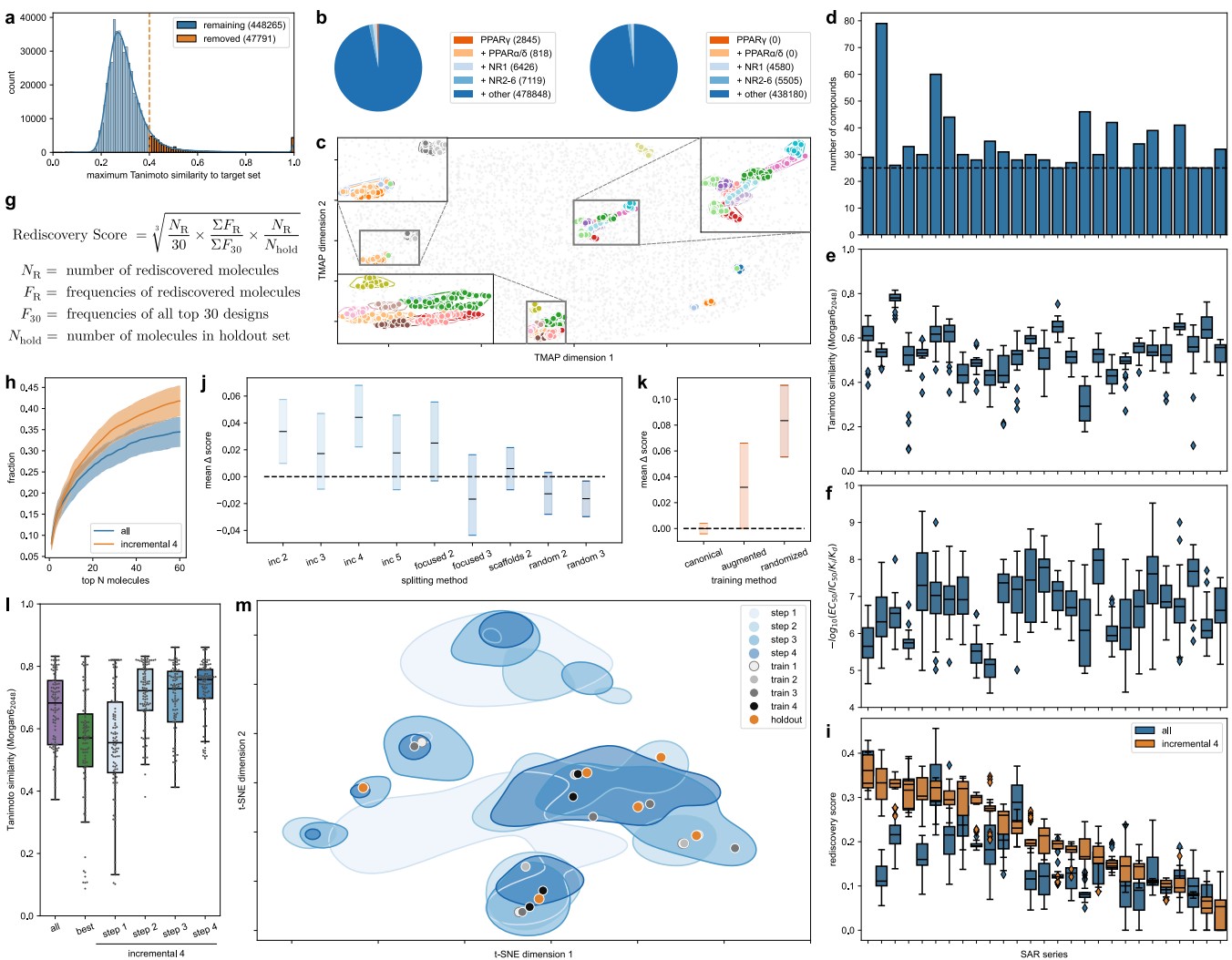

$$\text{Rediscovery Score} = \sqrt[3]{\frac{N_R}{30} \times \frac{\Sigma F_R}{\Sigma F_{30}} \times \frac{N_R}{N_{hold}}}$$

$N_R$ = number of rediscovered molecules
$F_R$ = frequencies of rediscovered molecules
$F_{30}$ = frequencies of all top 30 designs
$N_{hold}$ = number of molecules in holdout set

**Fig. 2 | Retrospective evaluation of incremental CLM training. a** A PPARγ ignorant CLM was developed using molecules from ChEMBL32 (bioactivity on any target ≤1 μM) excluding molecules with annotated activity on PPARs and molecules with Tanimoto similarity >0.4 (Morgan fingerprints, radius=2, 2048-bit). **b** Distribution of pretraining molecules with respect to activity on PPARs and related targets before (left) and after (right) activity-/similarity-based filtering. **c** TMAP[45] embedding of known PPARγ ligands revealed 27 SAR series (one color each). Background: 45k random ChEMBL molecules (5k shown). Characteristics of the 27 PPARγ agonist SAR series: number of molecules (**d**), potency distribution (**e**), internal pairwise Tanimoto similarity distribution (**f**). **g** Rediscovery score to evaluate CLM performance in structural optimization. **h** Fraction of rediscovered highly active ligands from the holdout set in the top-ranking designs retrieved by classical (all) or incremental CLM fine-tuning (n = 135; lines: mean, shades: SEM). **i** Rediscovery scores achieved by classical (all) and incremental fine-tuning for the 27 PPARγ agonist SAR series datasets (n = 25, five fine-tuning iterations with five sampling runs each). **j** Effects of different dataset splits on the rediscovery score.

The difference (Δ) in the rediscovery score to classical fine-tuning is shown. **k** Effects of data augmentation (×10) and epoch-wise SMILES randomization on the rediscovery score relative to the canonical model. **l** Tanimoto similarity (Morgan fingerprints[46]) of the top-100 designs (n = 100) by perplexity obtained from CLMs fine-tuned with all template molecules per SAR series (all), only the most active subset (best), or the incremental approach at the different steps, to the holdout set containing the most active compounds of each series. **m** Effect of incremental fine-tuning on design distribution visualized by t-distributed stochastic neighbor embedding (t-SNE) for one exemplary SAR series (regions were defined by the top-100 designs (perplexity) for each training step; background: 10k random ChEMBL molecules). Boxplots in **e**, **f**, **i**, **l** show median and interquartile range (IQR), whiskers extend to the most extreme points within 1.5 × IQR. Data in **h**, **j**, **k** from five fine-tuning iterations over 27 datasets (n = 135); black lines: mean Δ rediscovery score; shades: 95% CI derived from bootstrapping. Source data are provided as a Source Data file.

To mimic ligand design without previous knowledge and exclude any previous bias in the model, we trained a CLM that was ignorant of PPARγ ligands by removing all known modulators of the PPAR transcription factor family as well as all molecules with >0.4 Tanimoto similarity (Morgan fingerprints, radius = 2, 2048-bit) to a PPAR ligand from the pretraining dataset (Fig. 2a, b). The unfiltered pretraining data (molecules with at least one annotated bioactivity ≤1 μM on any human target from ChEMBL32[43]) contained 496,056 molecules (189,387 unique scaffolds) among which 2845 molecules had an annotated activity (no cutoff) on PPARγ and 818 additional molecules

were annotated as ligands of other PPAR subtypes. The activity- and similarity-based filtering excluded a total of 47,791 molecules representing 12,286 unique scaffolds and removed all PPARα/γ/δ ligands confirming effective elimination of knowledge on PPARγ from the pretraining (Fig. 2b, Supplementary Fig. 1). Comparison of CLMs developed with the full pretraining data, the PPAR-filtered set, or a further reduced dataset additionally lacking all ligands of the larger NR1 family that contains the PPARs (Supplementary Fig. 2) revealed higher perplexity of PPAR ligands for the model trained with the PPAR-filtered set confirming the intended ignorance of the target of interest.

In comparison, the further removal of all NR1 ligands from the pre-training data had a negligible additional effect on the perplexity of PPAR ligands but resulted in overall lower model performance as evident from greater deviation from the reference chemical space, quantified using Fréchet ChemNet Distance[44] (FCD) and Kullback-Leibler (KL) divergence across a set of chemical descriptors (Supplementary Fig. 2). These results corroborated the PPAR-filtered pre-training data (containing 448,265 molecules representing 177,101 unique scaffolds) as suitable to develop a PPARγ ignorant CLM.

To prepare datasets for focused fine-tuning on individual PPARγ ligand chemotypes and explore the ability of CLMs to optimize these scaffolds, the excluded known PPARγ ligands were grouped by similarity using TMAP[45] embedding to assign molecules to common chemotypes and SAR series (Fig. 2c). TMAP revealed 27 series of PPARγ ligands containing 25–79 individual molecules with 0.30–0.77 intra-set pairwise Tanimoto similarity computed on Morgan fingerprints[46] (radius = 3, 2048-bit) and spanning 1.3–4.3 orders of magnitude in potency (Fig. 2d–f) that appeared suitable to explore the potential of CLMs for structural optimization.

Using these PPARγ ligand series, we explored whether mimicking a medicinal chemist, who is learning incrementally from design-make-test cycles, could enable CLMs to discover the most active entities of each series based on limited fine-tuning data. For this, we split each PPARγ agonist series (i.e., chemotype) by potency for incremental CLM fine-tuning with increasingly active sets. The 25% most potent molecules of each series were kept as a holdout set and not used for fine-tuning to evaluate whether a CLM could design these target molecules after incremental fine-tuning with the remaining dataset. The discovery of higher-activity molecules from the holdout set—which are unknown to the model—can be a substantial challenge and would indicate that the CLM captured SAR features of the given scaffold and can effectively exploit the region of the chemical space containing the most active molecules.

As a measure to evaluate the success of this incremental fine-tuning with SAR data, we established a score (Fig. 2g) capturing the ability of the CLM to rediscover the holdout molecules. This rediscovery score is composed of three factors: the term ($N_R/30$) represents the number of rediscovered known highly active molecules in the 30 most probable designs based on perplexity[47]; the term ($\Sigma F_R/\Sigma F_{30}$) captures the sampling frequency of the rediscovered molecules in the 30 most probable designs in 2048 generated SMILES strings; and the fraction of rediscovered molecules of the entire holdout set is considered by ($N_R/N_{hold}$). The score hence captures, how many holdout molecules are discovered by the model in the top-ranking designs, how often the rediscovered holdout molecules are sampled, and how many holdout molecules are retrieved as fraction of the entire holdout set. The score was designed to compensate for extreme cases, i.e., the discovery of all holdout molecules at low frequency and the discovery of only a single holdout molecule at high frequency. Monitoring the score over exemplative fine-tuning procedures revealed that it could indeed balance extremes and highlight models with the desired performance of sampling many holdout molecules at high frequency (Supplementary Fig. 3).

Using the rediscovery score, we probed various fine-tuning modes by training the PPARγ ignorant CLM with molecules belonging to one SAR series. As illustrated by consistently higher rediscovery scores in almost all studied SAR series with incremental fine tuning, the CLM performance in designing potent PPARγ ligands (from the holdout sets) improved significantly with this approach (Fig. 2h, i; $p < 0.001$, linear mixed model effect by residual maximum likelihood). We observed an optimum for incremental learning with a four- to five-fold split of the SAR series for stepwise fine-tuning (Fig. 2j). Interestingly, data augmentation by using ten different SMILES to represent each fine tuning molecule was not as beneficial as in previous studies[13] and inferior to using one random SMILES representation for each fine

tuning molecule per epoch (Fig. 2k). Superior performance of the incremental approach was also evident from similarity evaluation of the 100 most probable designs ranked by perplexity to the holdout set containing the most active compounds per series which consistently increased over four sequential fine-tuning steps (Fig. 2l, m). For comparison, using all molecules of an SAR series or only the most active representatives (i.e., the last sequential set) in a single fine-tuning step led to substantially lower similarity and higher variance. The structural optimization performance of CLMs after incremental fine-tuning was independent of the size, internal diversity and potency range of the studied SAR series (Fig. 2d–f), indicating that it could be broadly applicable to structural optimization. These theoretical results thus highlighted great potential of CLMs for the challenging task of structural optimization of drug molecules for on-target potency and encouraged prospective application.

## A CLM designs optimized PPARγ agonists after incremental fine-tuning in prospective application

For prospective application to ligand optimization, we chose a series of benzimidazol-2-ylmethoxybenzene based PPARγ agonists from a cluster of four SAR studies[48–51] in the TMAP (Fig. 2c) spanning more than three orders of magnitude in potency (Fig. 3a, the full fine-tuning set details are shown in Supplementary Table 1 and in Supplementary Data 1) with **A**[48] as the most potent entity (Fig. 3a–c). No other known PPAR ligand was provided to the model. We used a five-step incremental fine-tuning with increasingly potent template sets (i.e., a fifth fine-tuning step instead of the holdout set) to focus a CLM on this scaffold. In contrast to the theoretical applications, the model was hence trained with all known molecules of the SAR series to explore the ability of CLMs to perform an evolutionary design step beyond the existing SAR knowledge. Similarity evaluation of the designs obtained from this model and comparison with the naïve CLM, fine-tuning with all templates (all) or only the most active templates (best) indicated that the incremental approach achieved a stronger focus towards the most active ligands in the template set (Fig. 3b) further supporting the design approach. We then generated 10k samples from the fully fine-tuned model and independently repeated the fine-tuning and sampling procedure five times. Of the resulting 50k designs, 99% were valid and 28% were unique. Among the top-ranking 1024 valid and unique designs according to perplexity[47], 799 (78%) represented the intended PPARγ agonist scaffold indicating that the CLM was strongly biased towards the chemical space of interest and that the design objective was met (Fig. 3d). We removed all known (annotated in ChEMBL or SureChEMBL) and highly similar molecules (≥0.8 Tanimoto similarity to known compounds, Morgan fingerprint, radius = 3, 2048-bit) as well as designs that were also obtained from the baseline CLM trained with all templates to prioritize incremental fine-tuning (Fig. 3e), and then selected designs for synthesis and testing by perplexity-based[47] ranking. The ten top-ranking compounds (**1-10**, Fig. 3f) displayed low differences in perplexity and resembled the intended PPARγ agonist chemotype as reflected by intermediate maximum similarity to the known fine-tuning molecules including the most active template **A** (Fig. 3g). Structural differences were mainly present in the hydrophobic backbone while the benzimidazole core and a carboxylic acid or a bioisosteric TZD motif were conserved indicating that the model captured the key SAR features for PPARγ activation. We engaged on **1-3**, and **5-10** for synthesis and testing. The computational designs were prepared over four (**1, 3, 7, 10**) to six (**2, 5, 6, 8, 9**) linear steps (Fig. 4a). The known PPARγ agonist **A** representing the most active entity of the SAR series used for fine-tuning (Fig. 3c, Supplementary Table 1 and Supplementary Data 1) according to the literature[48] was prepared for comparative testing in the same assay system.

In vitro testing demonstrated that all nine CLM designed compounds acted as highly potent PPARγ agonists with sub-nanomolar to

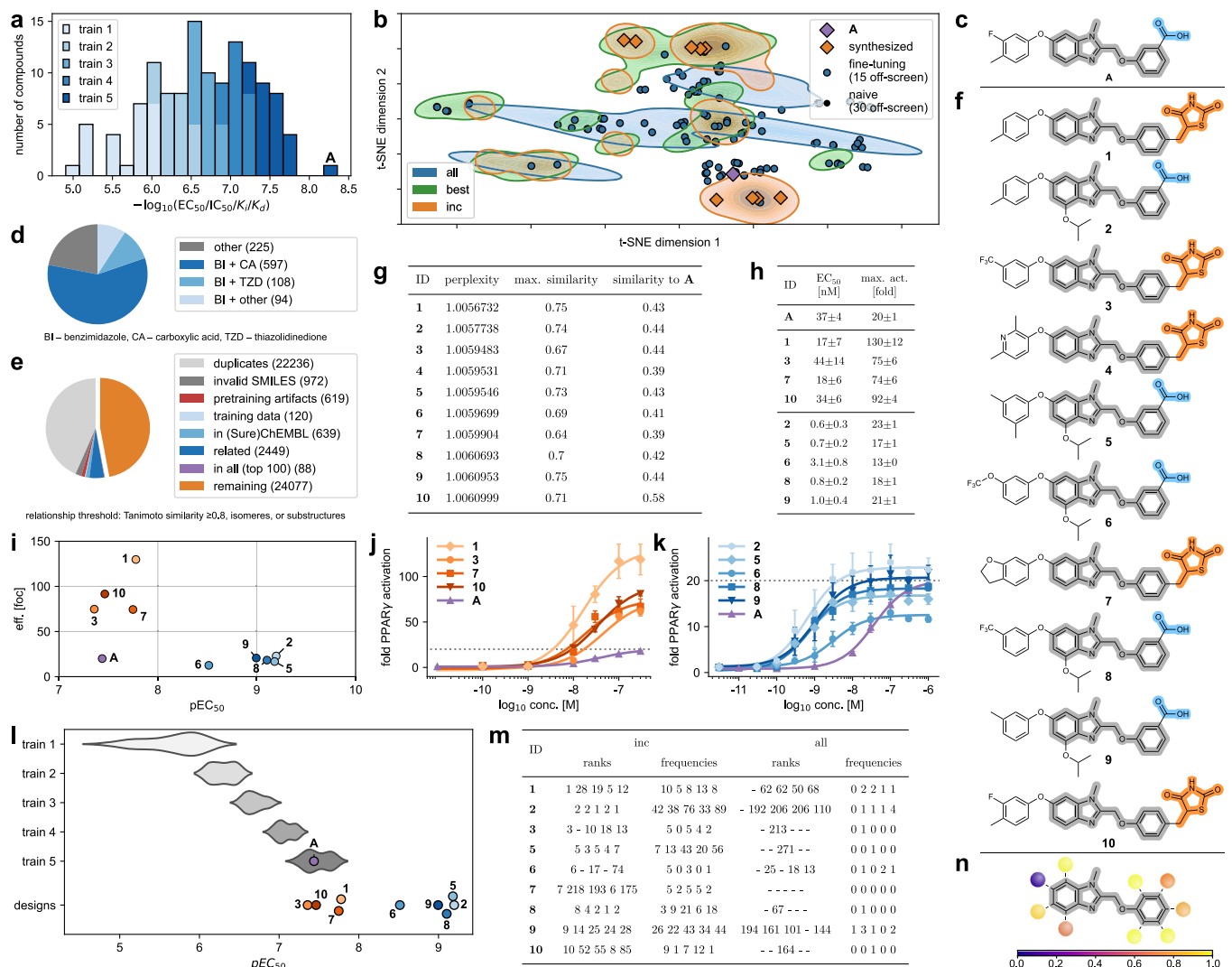

**Fig. 3 | Structural optimization of PPARγ agonists with incremental CLM training. a** Potency distribution of benzimidazole-based PPARγ agonists used for fine-tuning. Different colors represent the fine-tuning steps with increasing potency. **b** t-Distributed stochastic neighbor embedding (t-SNE; Morgan fingerprints[46]) of the top-30 designs by perplexity (regions) from a naïve CLM, after fine-tuning with all templates (all), only the most active templates (best), or the incremental approach (background: 10k random ChEMBL molecules). **c** Most active PPARγ agonist **A** of the fine-tuning series[48]. **d** Chemotype distribution of CLM designs (top-1024 valid and unique, by perplexity) after incremental fine-tuning. **e** Distribution of de novo designs (from five independent incremental fine-tuning iterations sampling 10k designs each) with respect to invalid SMILES, known and related (Tanimoto similarity ≥0.8, Morgan fingerprints, radius=3, 2048-bit; isomers; substructures) molecules, and designs also frequently sampled after classical fine-tuning (all). **f** Top-10 designs (**1**-**10**) by perplexity (after filtering (**e**)) from the incrementally fine-tuned CLM. **g** Perplexity and Tanimoto similarity of the top-10

designs to the most similar training molecule (cf. Supplementary Table 2) and the most active known agonist **A**. **h**, **i** Potency and efficacy of **1**-**3**, **5**-**10** and **A** on PPARγ. Dose-response curves of **1**, **3**, **7**, **10** and **A** (**j**) and **2**, **5**, **6**, **8**, **9**, **10** and **A** (**k**) in a PPARγ hybrid reporter gene assay. The dotted line shows the maximum eff. of **A**. Data in **h**–**k** are the mean ± SEM; *n* = 3 (three independent repeats in duplicates). **l** Potency distribution of the fine-tuning molecules[48–50,84] (train 1-5) and the CLM designs **1**-**3** and **5**-**10**. Retesting the most active fine-tuning entity **A** in the same setting as **1**-**3** and **5**-**10** revealed slightly lower potency than reported in literature. **m** Ranks (perplexity) and sampling frequencies of the designs **1**-**3** and **5**-**10** for incrementally fine-tuned CLMs (inc) and for CLMs trained with all template molecules in one go (all). Data from five independent fine-tuning and sampling procedures (10k designs each). **n** Scope of PPARγ agonist scaffold modifications designed by the incrementally fine-tuned CLM (darker colors represent higher diversity). Source data are provided as a Source Data file.

low nanomolar EC$_{50}$ values (Fig. 3h–l) indicating that the CLM captured SAR features driving high potency on the target of interest by incremental fine-tuning. The four TZD-containing designs **1**, **3**, **7** and **10** outmatched the known template agonist **A**[48] in PPARγ activation efficacy by up to a factor of 6.5. **1** and **7** additionally displayed slightly superior potencies compared to **A**. **1**, **7** and **10** caused stronger PPARγ activation at 10 nM than the reference compound **A** at its maximum plateau (dotted line in Fig. 3j), which is only reached above 100 nM, underscoring the enhanced biological activity of these CLM designs compared to the fine-tuning molecules. The benzoic acid derivatives **2**, **5**, **6**, **8** and **9** exhibited even more striking PPARγ agonism with EC$_{50}$

values of 0.6 to 3.1 nM corresponding to 12- to 62-fold increased potency compared to the reference **A** (EC$_{50}$ 37 nM). With nine out of nine tested CLM designs exceeding the most active known template in PPARγ agonist activity, these results impressively confirmed the ability of incrementally fine-tuned CLMs to extend SAR knowledge used for training and to design more potent analogs.

The optimized designs **1**-**3**, **5**-**10** were consistently sampled with high frequency and at high perplexity-based ranks when the incremental fine-tuning procedure was independently repeated (Fig. 3m), but not present in the top-ranking designs of a CLM trained with all SAR data for the given PPARγ agonist scaffold in one

**Fig. 4 | Synthesis of 1-3, 5-10, 27-29, A and C. a** Synthesis of PPARγ agonists designs **1**-**3** and **5**-**10**, and the most potent agonist **A** from the training data; reagents and conditions: (i) NaH, DMF, 90 °C, 3 h, 79–93%; (ii) iron powder, conc. HCl, MeOH/H₂O (9:1), 90 °C, 2 h, 13–16%, or iron(III)acetylacetonate, hydrazine hydrate, DMF, 80 °C, o.n., 20–33%; (iii) thiazolidine-2,4-dione, KOH, EtOH, rt, o.n., 99%, (iv) CoCl₂-DMG complex, NaBH₄, NaOH, H₂O/MeOH (3:1), 35 °C, 3 h, 40%; (v) **14a-d, 17**, DIPEA/HATU, DMF, 3 h, r.t., then 3 h, 90 °C, 10–19%; (vi) *tert*-butyl bromoacetate, K₂CO₃, DMF, rt, o.n., then TFA/CH₂Cl₂ (1:1), anisole, rt, 3 d, 43%; (vii) **14d**, DIPEA/HATU, DMF, 3 h, r.t. and 3 h, 90 °C, 17%; (viii) NaOH/1,4-dioxane (2:1), 2 h, 80 °C; quant.; (ix) isopropanol, PPh₃, DBAD, THF, 0 °C, 1 h, then rt, o.n., 74%; (x) MeNH₂, THF, 40 °C,

o.n., 99%; (xi) NaH, DMF, 80 °C, o.n., 35–96%; (xii) Fe, NH₄Cl, EtOH/H₂O, 80 °C, o.n.; (xiii) TCFH, NMI, MeCN, 0 °C, 1 h, then r.t., o.n., 7–72% over 2 steps; (xiv) 1 N HCl in 1,4-dioxane, 80 °C, o.n., then NaOH(aq), 80 °C, 2 h, 48–97%. **b** Synthesis of RORγ ligand designs **27-29**, and the most potent and the most active entity **C** from the training data; reagents and conditions: (xv) NEt₃ or DIPEA, CH₂Cl₂, 0 °C, 2–21 h, 16–63%; (xvi) NaH, *N,N*-dimethylacetamide, r.t. to 80 °C, 16 h, 29–73%; (xvii) Pd₂(dba)₃, XPhos, Cs₂CO₃, toluene, 115 °C, 21 h, 39-65%. (xviii) TEA, CH₂Cl₂, rt, 16 h, 47–70%; (xix) NaH, DMSO, rt, 1 h, 67–91%; (xx) NaOtBu, XPhos Pd G3, 1,4-dioxane, 80 °C, 2 h, 18-25%.

go (all), or the naïve CLM. This design preference of the incrementally trained model indicated that the CLM better captured the chemical features required for potent PPARγ agonism from stepwise fine-tuning boosting its ability to design optimized descendants based on the available SAR information. This was also evident from analyzing the structural differences in the sampled designs which centered at few substituent positions (Fig. 3n) and displayed the highest diversity for the hydrophobic group in 6-position of the benzimidazole scaffold. The structural variations aligned well with the known SAR within the fine-tuning set and of related molecules outside the cluster[52,53] (not used for fine-tuning). Compared to the most active fine-tuning template **A**, the carboxylic acid was frequently replaced by a methylthiazolidinedione motif in top-ranking designs (**1**, **3**, **4**, **7**, **10**) which enables improved interaction with the PPARγ activation function[54]—although this information was not explicitly available to the model. Simultaneously, the CLM adapted the regiochemistry (from *meta*- to *para*-position) likely improving the orientation of the acidic motif in the binding site. In the benzoic acid derivatives (**2**, **5**, **6**, **8**, **9**), the model introduced an iso-propyloxy substituent on the benzimidazole core—a modification that is also present in a subset of the fine-tuning molecules and boosted potency of the optimized designs. These results suggest that the model learned key SAR features and their interdependencies from the structured SAR data used for fine-tuning, which enabled the de novo design of optimized analogs outmatching the fine-tuning molecules (Fig. 3h–l).

## A CLM designs highly potent inverse RORγ agonists

After successful application to structural optimization of PPARγ agonists, we employed the incremental CLM training approach to explore the SAR of inverse RORγ agonists as further example target. This receptor is an integral part of the circadian clock and a key regulator of lymphocyte differentiation[55,56], and is receiving attention in drug discovery for inflammatory and auto-immune diseases with several inverse RORγ agonists being studied in clinical trials[56,57]. As for PPAR, we developed a CLM ignorant of ROR ligands (Fig. 5a, Supplementary Figs. 4, 5) and identified ten inverse agonist clusters (Fig. 5b) with broad SAR knowledge in literature for incremental CLM fine-tuning. Compared to PPARγ, we employed larger (78–132 molecules) and chemically more diverse (internal Tanimoto similarity 0.25–0.47) SAR series. Nevertheless, retrospective evaluation using the rediscovery score supported the CLM's ability to capture SAR data from the incremental training approach and to design increasingly potent inverse RORγ agonists (Fig. 5c), indicating that the incremental approach could also generalize within a larger target chemical space. For prospective application, we chose a series of arylsulfonamide-based RORγ modulators[58–61] (full fine-tuning set details are shown in Supplementary Table 3 and in Supplementary Data 1) and sampled designs after incremental fine-tuning with this chemotype. Designs with a similarity >0.6 to a fine-tuning entity were removed to account for higher diversity of the SAR series and sulfonamides were prioritized. From the top-10 remaining most frequently sampled designs, we prepared **27** and **28** (Fig. 5d) via three synthetic steps (Fig. 4b) for in

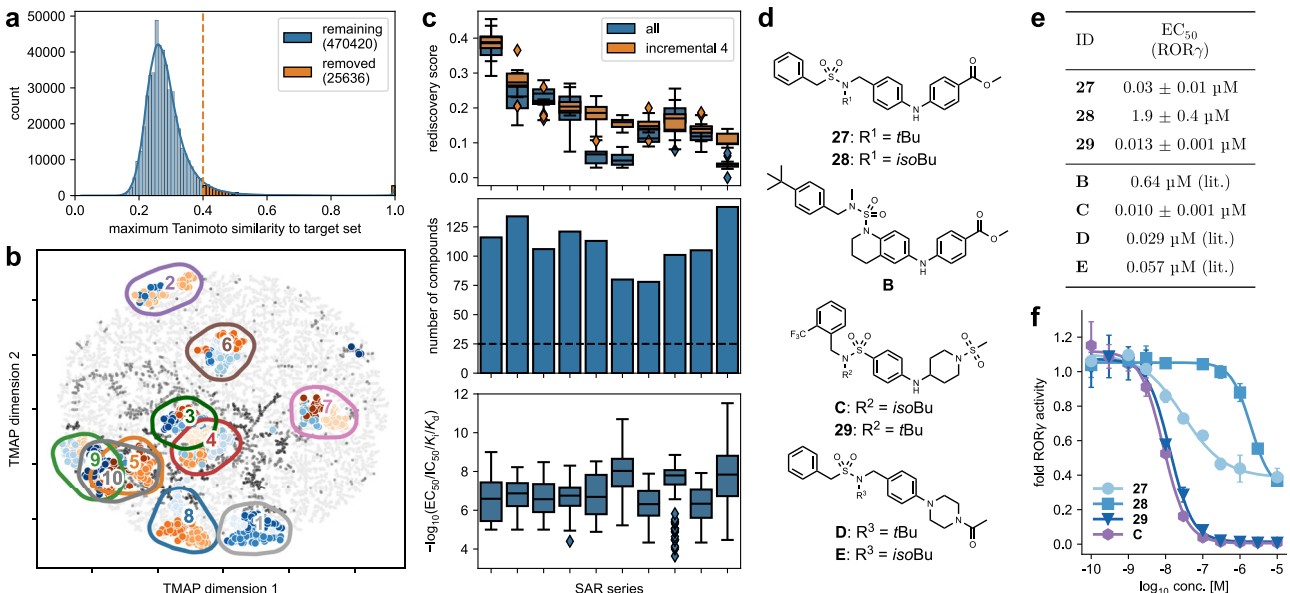

**Fig. 5 | Design of inverse RORγ agonists with an incrementally fine-tuned CLM.**
**a** A CLM ignorant of ROR ligands was developed using molecules from ChEMBL32 from which all known ROR ligands as well as all molecules with Tanimoto similarity >0.4 (Morgan fingerprints, radius = 2, 2048-bit) to a known ROR ligand were removed. **b** Known RORγ ligands were clustered using TMAP[45] embedding to identify related chemotypes (ten in total, each represented by one color). The background chemical space was calculated using 10k random ChEMBL molecules. **c** Rediscovery score from retrospective evaluation (*n* = 25, five fine-tuning iterations with five sampling runs each), number of contained molecules, and potency distribution of the ten inverse RORγ agonist chemotype datasets (*n* = number of molecules in respective dataset). Box plots display median and IQR, with whiskers

extending to extreme values within 1.5 × IQR; **d** Chemical structures of designs **27** and **28** from a CLM incrementally fine-tuned with a cluster of inverse RORγ agonists[58–61] (Supplementary Table 3), the most similar (**B**[58]) and most potent (**C**[61]) inverse RORγ agonists from the corresponding fine-tuning data, the (human-designed) analog **29** of **C**, the only *N-tert*-butylsulfonamide **D** in the training data and its *N-sec*-butyl equivalent **E**. The most similar fine-tuning molecules to **27-29** are shown in Supplementary Table 4. Potency, efficacy (**e**) and dose-response curves (**f**) of **27-29** and **C** in a Gal4-RORγ hybrid reporter gene assay. Data are the mean ± SEM, *n* = 3 (three independent repeats in duplicates). Data for **B, D**, and **E** from lit[58,60]. Source data are provided as a Source Data file.

vitro testing. **27** and **28** resembled the chosen RORγ modulator chemotype but differed in the arrangement of the scaffold's pharmacophore elements and displayed only intermediate Tanimoto similarity on functional-connectivity fingerprints (FCFP) (0.556 and 0.578) to the most similar fine-tuning molecule **B** (Supplementary Tables 3, 4). Evaluation of RORγ modulation (Fig. 5e, f) demonstrated high potency for **27** markedly (20x) exceeding the most similar fine-tuning entity **B**. **27** was at least 3-fold more active as inverse RORγ agonist than any compound of the SAR series of **B**[58] and almost matched the most potent compound (**C**[61], Fig. 5d, Supplementary Table 3) in the cluster but displayed rather high chemical diversity to **C** (Tanimoto similarity 0.302).

Although the computational designs **27** and **28** differ only in their *N*-substituent on the sulfonamide motif, **28** was markedly less active as inverse RORγ agonist demonstrating strong impact of this substructure on potency. The highly active design **27** comprises an *N-tert*-butylsulfonamide group which is replaced by *sec*-butyl in **28** resulting in a major drop in inverse RORγ agonism. Although the *N-tert*-butyl substructure was only present in the fine-tuning data in one molecule contained in the highest potency set (*sec*-butyl in 23 fine-tuning templates), the CLM nevertheless prioritized this less frequent but more potent moiety. To test whether the *N-tert*-butyl group was generally driving potency or had a context-specific impact, we transferred it to the most active compound **C** in the cluster and prepared the corresponding *tert*-butyl analog **29**. Of note, **29** was not sampled as a high-ranking design by the CLM but human-designed to study relevance of the *N*-substituent. **29** displayed equal potency as **C** demonstrating that the *tert*-butyl group was tolerated but enhanced potency only in a context- and scaffold-specific fashion. Indeed, the only *N-tert*-butyl containing fine-tuning molecule **D** (Fig. 5d, Supplementary Table 3) resembles **27** and **28** by comprising the sulfonamide in an *N*-benzyl-1-

phenylmethanesulfonamide substructure indicating that the incrementally fine-tuned CLM captured long-range structural dependencies in the SAR of the scaffold.

## Incremental fine-tuning improves model perplexity and SAR perception

The observed performance in structural optimization of drug molecules in retrospective and prospective applications highlighted incremental CLM fine-tuning with increasingly potent templates as superior approach for biasing the models towards the chemical space of highly active ligands of a given target of interest. Possible explanations for this improved performance lie in an improved learning trajectory of the model and the added dimension in the structured training data. Successively providing the CLM with more active templates resembling a semi-ranked list may create a smoother gradient landscape during fine-tuning allowing the model to reuse and refine previously learned knowledge efficiently. Additionally, the time-step-like fine-tuning could match well with the ability of recurrent neural networks to capture sequence-based data[4].

To explore whether incremental fine-tuning also had a tangible impact on the models, we compared the effects of different training regimen on design perplexity as internal performance measure. The perplexity of target molecules from the holdout set indeed developed differently for different fine-tuning strategies (Fig. 6a). Using all or only the most active templates (best) in a single fine-tuning dataset resulted in a continuous perplexity reduction until a steady state was reached. Incremental fine-tuning, in contrast, revealed a stepwise perplexity improvement consistent with the stepwise availability of increasingly potent training molecules. After the full fine-tuning procedure, the perplexity of the highly active holdout set entities was significantly lower with incremental fine-tuning (*p* < 0.001; linear mixed model

effect; over all 27 PPARγ SAR datasets with five repeats each) compared to the use of all templates although the same overall training data were used. Compared to using only the most potent fine-tuning subset, the superior performance of the incremental approach was even more striking ($p < 10^{-5}$).

Improved perception of SAR data by incrementally trained CLMs was also evident in correlation analysis between potency and perplexity for the traditional (all) and incremental fine-tuning strategies (Fig. 6b). When the model was fine-tuned with all templates in one go, perplexity of the PPARγ ligands including the holdout set displayed low variance indicating that the model considered all molecules of the series equally and could not distinguish low and high potency. In some cases, improved perplexity and higher potency were even negatively correlated possibly due to higher frequency of low-potency entities in the training data. Incremental fine-tuning, in contrast, despite using the same overall data consistently produced positive correlations between improved perplexity and higher potency for the entire dataset including the unknown holdout molecules. This impact of incremental fine-tuning on the perplexity-potency correlation further supported that the models could better capture potency-driving SAR features from the stepwise training and more effectively exploit the chemical sub-space of high-activity molecules.

## Discussion

Structural optimization of a given drug scaffold for enhanced activity on a given target in hit-to-lead campaigns in drug discovery is costly and time-consuming. While many computational techniques have been implemented to aid this task, most generative design approaches focused on simpler molecular properties (e.g., logP, QED) than bioactivity and omitted the on-target potency optimization challenge. Additionally, generative structural optimization models rely on external scoring (e.g., docking), or leverage information from the target protein or matched molecular pairs. Structural optimization for on-target potency purely based on ligand information from SAR knowledge and without external scoring has not been established. We closed this gap by exploiting the documented strength of CLMs in designing new molecules with desired properties based on task-specific fine-tuning. We capitalized on the ability of recurrent neural networks to capture sequential data and added another dimension to the fine-tuning by providing the model with increasingly potent templates in consecutive steps creating a learning trajectory over the SAR data. As evident from a higher rediscovery rate of highly active holdout molecules and stronger correlation between perplexity and potency of molecules representing the given chemotype, the CLMs could learn better from this incremental fine-tuning approach. This beneficial effect on CLM performance may emerge from a smoother gradient landscape during fine-tuning as domain shifts in the optimization trajectory from the pretrained towards the fully fine-tuned model are attenuated. Additionally, the stepwise presentation of ranked templates to the model may match well with the ability of LSTM to capture sequential (time-step)[4] data. Perplexity monitoring over different fine-tuning strategies indeed suggested that the model learned in a stepwise manner from incremental fine-tuning and eventually achieved superior perplexity-potency correlation for the training and the holdout molecules using this approach.

The prospective results impressively confirmed the potential of CLMs to capture existing SAR data and extend this knowledge towards novel, more active analogs highlighting their applicability not only to the de novo design of new ligand scaffolds for a given biological target but also for driving structural optimization of existing chemotypes. Therein, the incremental fine-tuning approach intentionally promotes chemical similarity of designs and templates which is required to achieve the task of structural optimization within a given series of bioactive molecules.

In the exemplative prospective application on PPARγ agonists, stepwise fine-tuning with SAR data focused on a single bioactive scaffold successfully biased the model for the desired chemotype and enabled effective exploration of this narrow chemical space to design close analogs exceeding the most active entity of the fine-tuning series in PPARγ agonism. Structural comparison of designs and templates revealed that the model captured the given SAR and prioritized favored motifs. In a fraction of the top-ranking CLM designs, the carboxylic acid group found in the highest potency template **A** was replaced by a methylthiazolidinedione resulting in markedly increased agonist efficacy. The TZD group is known to interact stronger with the PPARγ activation function, but this information was not explicitly available to the model. Notably, the carboxylate was not simply replaced by the bioisoster but the CLM simultaneously adapted the regiochemistry. The remaining top-10 designs were extended by an additional iso-propyloxy substituent compared to **A**. This structural modification was extracted by the CLM from the highest potency fine-tuning subsets and resulted in markedly increased PPARγ agonist potency of five CLM designs over **A**.

In the prospective design of RORγ modulators, our results demonstrated that incremental fine-tuning can also be applied to larger and more diverse compound clusters which enabled a transfer of SAR knowledge between (related) scaffolds. Therein, the CLM revealed an activity cliff between **27** and **28** and extracted the potency-driving *N-tert*-butylsulfonamide motif (**27**) despite its low frequency among RORγ ligands and in the fine-tuning data. Interestingly, the beneficial impact of this motif was context-/scaffold-dependent as evident from the lack of increased potency for the human-designed *N-tert*-butyl analog **29** of **C**.

The results of both prospective applications suggest that the CLMs not only captured key potency-driving SAR features but also their long-range structural interdependence within the SAR. Both scenarios, the focused analogue optimization as shown for PPARγ, and the broader SAR transfer exploration as demonstrated for RORγ, are highly relevant for structural optimization in drug design. The CLM framework for incremental fine-tuning (available at https://github.com/AkMerk/incremental-clm) may hence be valuable in other structural optimization scenarios. Future users should consider that the structured fine-tuning datasets (size, diversity, data split) require careful adaptation to the respective project and design objectives.

The impressive performance in SAR extension and the de novo design of highly potent analogs corroborates incrementally fine-tuned CLMs for structural optimization of drug molecules and extends the scope of CLMs in molecular design.

## Methods

### Software
Data were collected using Python (v3.10.7) with Pandas[62] (v2.2.1), and Polars[63] (v1.29.0). Data analysis was performed using Python (v3.10.7) with RDKit[64] (v2023.05) and tensorflow[65] (v2.11.1). SciPy (v1.15.3)[66], scikit-learn[67] (v1.4.2), and statsmodels[68] (v0.14.5) were used for statistical analysis. R[69] (v4.4.0) was used with lme4 (v1.1) and lmerTest[70] (v3.1-3) for linear mixed-effects modeling. Visualizations were generated using TMAP[71] (v1.0.18), matplotlib[72] (v3.8.3) and seaborn[73] (v0.13.2).

### Data processing
Molecules were encoded as canonical SMILES using RDKit[64] (v2023.05, www.rdkit.org) and standardized in Python (v3.10.7, www.python.org) by removing isotopes, stereochemistry, salts, duplicates, and neutralizable charges. Molecules with a molecular weight greater than 1000 Da were excluded. The SMILES were tokenized, and sequences exceeding 140 tokens were removed from the dataset.

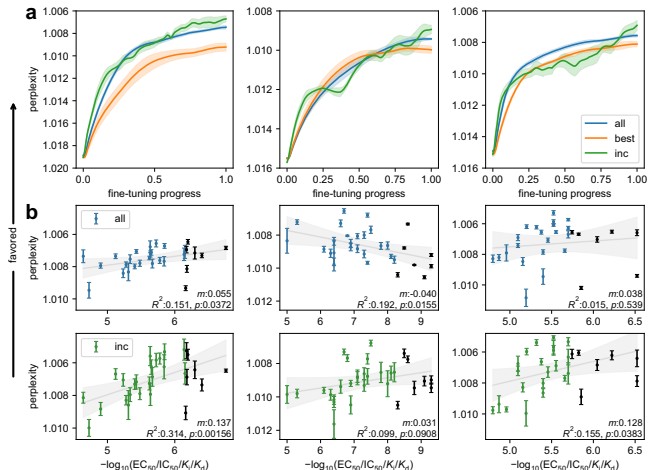

**Fig. 6 | Incremental fine-tuning improved model perplexity for high-potency molecules and overall potency-perplexity correlation.** Results are shown for three exemplative datasets for fine-tuning with all or only the most active (best) templates in one go, or the incremental approach. **a** Evolution of holdout set perplexity across the training process for three datasets using all (blue), best (orange), and incremental (green) fine-tuning. The fine-tuning progress is scaled (0–1) by epochs; Lines are the mean, shades represent SEM of five repeats per dataset. **b** Perplexity values of training and holdout molecules at the final epoch of all or incremental fine-tuning in relation to their $-\log_{10}$ potency. Log-linear correlations are displayed with regression lines and 95% CI of the fitted mean (gray line and shading). Perplexity markers denote mean ± SEM over five repeats. The y-axis is inverted to visualize a positive correlation between improved perplexity scores and potency, black dots are the holdout molecules (not seen by the models). For each correlation, the regression slope ($m$, scaled by 100), $p$-value (two-sided $t$-test), and correlation coefficient ($R^2$) are annotated. Source data are provided as a Source Data file.

## Pretraining datasets

Molecules were obtained from ChEMBL[43] (v32), restricted to compounds with at least one annotated bioactivity ≤1 μM on any human target. To construct PPARγ and RORγ ignorant datasets, we excluded all compounds with any assay annotation against human PPAR or ROR receptors (including their respective transcription factor families). In addition, we removed all compounds with a Tanimoto similarity >0.4 (Morgan fingerprint, radius = 2, 2048-bit) to any of the excluded ligands. The remaining data were randomly split into training (80%), validation (10%), and test (10%) sets. The training set was used for model pretraining, the validation set for epoch selection, and the test set was reserved for performance evaluation.

## Fine-tuning sets

For PPARγ, compound-bioactivity data points were collected from unique publications in ChEMBL[43] and BindingDB[74] (v2023.06), defined by their PubMed ID. Only functional assays reporting agonistic activity were included. SAR series with a mean internal Tanimoto similarity below 0.3 were excluded, along with single compounds having a Tanimoto similarity below 0.5 to their respective series. SAR series containing fewer than 25 compounds were also removed, resulting in 27 SAR series for retrospective analysis. For prospective application, a cluster of 4 SAR series remote from others in the TMAP projection was selected. For RORγ, all ligands were obtained from ChEMBL (v32). Agonists and allosteric binders were excluded. TMAP embedding was applied to cluster SAR series from various sources sharing the same or similar scaffolds. Ten clusters were selected for retrospective analysis, and one cluster was used for prospective application.

## Data splitting

For *all*-data fine-tuning, the entire PPARγ ligand dataset was used without splitting. For *values*-based splitting, the data was divided into $N$ equally sized groups with successively increasing potency. In the *focused* split, the data was split so that each subsequent group contained half the size of the previous one, also ordered by potency. For the *scaffold*-based strategy, fine-tuning was first performed on Murcko scaffolds[75], followed by fine-tuning with the complete SMILES strings. The *random* split strategy involved shuffling the data into training sets. The *best* split only used the set with the most active compounds.

## CLM implementation and pretraining

CLMs were implemented in Python (v3.10.7) using TensorFlow[64] (v2.11.1) and the Keras API (www.keras.io). The model architecture for PPARγ included five layers totaling 5,789,297 parameters: layer 1: Embedding (64 dimensions), layer 2: BatchNormalization, layer 3: LSTM (1024 units), layer 4: LSTM (256 units), layer 5: BatchNormalization, layer 6: Dense (64 units). The CLMs for RORγ ligands featured three LSTM layers with 1024 units each. The CLMs were trained on SMILES strings using the Adam optimizer[76] (initial learning rate = $10^{-3}$) and categorical cross-entropy loss. Three training versions were used: Canonical (only SMILES processed by RDKit's canonicalization algorithm), augmented[77] (SMILES augmented 10 times), and randomized[78] (SMILES were randomized after each epoch). Pretraining was conducted for 40 epochs (canonical), 20 epochs (augmented), and 100 epochs (randomized). Augmented and canonical training included learning rate reduction on plateau (factor 0.5, patience 3, until $10^{-4}$). The epoch used as the naïve CLM was chosen based on sampling statistics and distributional similarity, evaluated on a set of 10,000 generated SMILES compared with a random subset of 10,000 SMILES from the validation set (cf. Sampling statistics, Distributional similarity). A dropout of 0.2 was applied to each LSTM layer for the canonical and augmented versions. The CLMs for RORγ ligands were trained using both randomized and canonical training versions.

## Incremental fine-tuning

The naïve CLMs were fine-tuned for 100 epochs per step (i.e., fine-tuning subset), starting with a learning rate of $10^{-4}$. Canonical and augmented versions applied learning rate reduction on plateau (cf. Pretraining). 10% of the SMILES at each step were reserved as validation set. After fine-tuning, the model weights and optimizer state were reset to the epoch with the minimum validation loss (for randomized training, a moving average across 10 epochs). Training then continued with these parameters into the next step with the next fine-tuning subset.

## Sampling statistics

We quantified the validity, uniqueness, and novelty of the generated molecules. Validity was defined as the ratio between the number of RDKit[64]-parsable SMILES strings (v2023.05) and the total number of generated samples. Uniqueness was calculated as the proportion of unique SMILES among all valid molecules. Novelty was defined as the fraction of unique molecules that were not present in the training set.

## Distributional similarity

Distributional similarity metrics were implemented following the GuacaMol benchmarking framework[69]. We computed the FCD[44] using the reference implementation (v1.2.2) and transformed into a similarity score using

$$S_{\text{FCD}} = e^{-0.2 \times \text{FCD}} \tag{1}$$

In addition, we computed the Kullback−Leibler (KL) divergence[79] between generated and reference distributions across maximum internal Tanimoto similarity (Morgan fingerprints, radius = 2, 2048-bit) as well as a set of physicochemical descriptors: Bertz complexity

index[80], logP, molecular weight, total polar surface area, number of hydrogen-bond acceptors, number of hydrogen-bond donors, number of rotatable bonds, number of aliphatic rings, and number of aromatic rings. Continuous descriptors were modeled using Gaussian kernel density estimation as implemented in SciPy[66] (v1.15.3), while discrete descriptors were compared using 10-bin histograms. The resulting KL divergences were aggregated into similarity scores as

$$S_{\mathrm{KL}} = \frac{1}{D} \sum_d e^{-\mathrm{KL}_d} \qquad (2)$$

where $\mathrm{KL}_d$ denotes the KL divergence for descriptor $d$, and $D$ is the total number of descriptors. All molecular descriptors and fingerprints were calculated using RDKit[64] (v2023.05).

## Temperature/multinomial sampling

SMILES were sampled up to 140 characters using the softmax function parameterized by a sampling temperature of 1.0. The probability of the $i$-th character to be sampled from a CLM was computed as:

$$q_i = \frac{\exp\left(\frac{z_i}{T}\right)}{\sum_j \exp\left(\frac{z_j}{T}\right)} \qquad (3)$$

where $z_i$ is the CLM prediction for character $i$, $T$ is the temperature, and $q_i$ is the sampling probability of character $i$.

## Perplexity ranking

Invalid SMILES, duplicates, and SMILES present in the fine-tuning set were removed. The remaining SMILES were ranked by perplexity[47]. For the augmented and randomized version, each canonical SMILES was augmented 10 times, and the perplexity scores were averaged. Perplexity (*PPL*) was calculated as:

$$\mathrm{PPL} = \exp\left(-\frac{1}{N} \sum_{i=1}^{N} \log(q_i)\right) \qquad (4)$$

where $N$ is the number of characters in the SMILES and $q_i$ is the sampling probability of character $i$.

## Retrospective evaluation

Prior to splitting and training, 25% of the dataset was withheld as a holdout set. The naïve model was fine-tuned five times for each dataset and splitting strategy. Due to its non-determinism, the random splitting process was repeated three times, resulting in three distinct splits. After the final iteration of each training, 2048 SMILES were sampled and ranked by perplexity. SMILES with a naïve model perplexity below 1.012 were removed, and the top-ranking 30 designs were compared with the holdout set and the rediscovery score calculated for each method (cf. Fig. 2). Sampling and scoring were repeated five times for each fine-tuned model.

## Prospective application

For prospective application, the holdout set was appended as an additional training step, resulting in a 5-step (PPARγ) or 3-step (RORγ) *value*-based split, along with the respective naïve CLM and training strategies (cf. CLM pretraining and fine-tuning). 10,240 designs were sampled per run from five full fine-tuning repeats, combined, and filtered to remove duplicates and invalid SMILES strings. Molecules present in the training data, known in ChEMBL[43] (v32), or SureChEMBL[81] (v2024.01.01) were excluded, as were those appearing in the top-ranking 100 filtered results generated using a CLM classically fine-tuned with all data. Additionally, molecules that were substructures or isomers of training molecules or displayed high Tanimoto similarity (>0.8 for PPARγ, >0.6 for RORγ) were removed. To

remove potential artifacts introduced during the pretraining phase, PPARγ agonist designs with a naïve perplexity below 1.012 were discarded (Supplementary Fig. 6). From the RORγ ligand designs, molecules comprising unfavorable substructures (1,3-cyclohexadiene, rings larger than seven atoms) were removed, and sulfonamides were prioritized according to the fine-tuning chemotype. The top-ranking 10 designs (ranked by perplexity for PPARγ or abundance for RORγ) were considered for synthesis and testing.

## TMAP

The TMAP embedding[71] was performed to visualize the chemical space of the target ligands within a context of random ChEMBL molecules (PPARγ: 44,755, RORγ: 10,000). MinHash fingerprints (MHFP)[82] were computed with 1024 dimensions, a maximum radius of 3, and a minimum radius of 0. For PPARγ, only the filtered ligands were included. Both TMAP and MHFP calculations were performed using the framework available at https://github.com/reymond-group/tmap.

## Stochastic neighbor embedding

The t-SNE projection was performed with scikit-learn[67] (v1.4.2) in Python (v3.10.7) based on a Truncated SVD[83] to reduce Morgan fingerprints to 50 dimensions (distance metric: Tanimoto) using a background of 10,000 random ChEMBL molecules. Perplexity was set to 30, while all other parameters were used at their default values. For spatial visualization, the Gaussian kernel density estimation was applied using seaborn[73] (v0.13.2) with the bandwidth adjustment set to 0.4 for analyses based on 30 samples per group and to 0.5 for analyses based on 100 samples per group.

## Statistical analysis

Ordinary linear regressions were performed as log-linear models, with the negative decadic logarithm of activity regressed against the mean perplexity, using statsmodels[68] (v0.14.5) in Python (v3.10.7). Linear mixed-effects models were calculated in R[69] (v4.3.3) using lmerTest[70] (v3.1-3). The models included either the mean rediscovery score or the mean holdout perplexity as fixed effects, with the dataset included as a random effect to account for variability between datasets.

*A code summary (pseudocode) is shown in* Supplementary Fig. 7. *Synthetic procedures, analytical characterization, and in vitro assay methods are described in the Supplementary Information.*

## Reporting summary

Further information on research design is available in the Nature Portfolio Reporting Summary linked to this article.

## Data availability

All data supporting the findings of this study are available within the paper and its Supplementary Information. Source data are provided with this paper. The raw data used to pretrain chemical language models were retrieved from ChEMBL (v32); the raw data used for fine-tuning were retrieved from ChEMBL and BindingDB (v2023.06). The processed data used for pretraining and fine-tuning are available on GitHub at https://github.com/AkMerk/incremental-clm/ (https://doi.org/10.5281/zenodo.18773078), in the Supporting Information (Supplementary Data 1) and are available upon request. Source data are provided with this paper.

## Code availability

Code used in this study is available at https://github.com/AkMerk/incremental-clm/. (https://doi.org/10.5281/zenodo.18773078).

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

## Acknowledgements

This research was funded by the European Union (ERC, NeuRoPROBE, 101040355, to D.Me.). Views and opinions expressed are however those of the author(s) only and do not necessarily reflect those of the European Union or the European Research Council. Neither the European Union nor the granting authority can be held responsible for them.

## Author contributions

T.H. prepared the code and trained, evaluated and applied the models with contributions from A.H. and T.W.; D.Ma. and M.L. prepared and tested the computational designs. T.H. prepared the figures. D.Me. conceived the study, supervised the project and wrote the manuscript with contributions from all authors.

## Funding

## Competing interests

We declare that none of the authors have competing financial or non-financial interests as defined by Nature Portfolio.
