## [Transparent Peer Review file · Nature Communications]

Structural optimization of drug molecules with incrementally trained language models

Corresponding Author: Professor Daniel Merk

Version 0:

Reviewer comments:

Reviewer #1

(Remarks to the Author)

The manuscript "Structural optimization of drug molecules by generative deep learning" has become significantly clearer because of the changes and additions made. Below, a point-by-point reflection is provided on the initial concerns raised in light of the amended manuscript.

-Comment on the setting with regards to previous work: The authors have strongly improved the introductory setting of their manuscript, which now allows the reader to appreciate their work much better.

-Comment on the CLM approach: additions to the manuscript have provided more clarity for the choice of LSTM by the authors.

-Comment on the PPARg ligands: The textual clarifications added to the manuscript help in appreciating the approach taken by the authors and in appreciating the data. Notwithstanding, the selection of compounds 1, 3, 7, and 10 for synthesis and testing (out of the ten top-ranking compounds) gives the reader a sense of (strong) bias: these are all TZDs, while none of the carboxylic acids (2, 5, 6, 8, 9), which were similarly ranked in the top ten, were selected for synthesis and testing. Even though the selection of only the TZDs might potentially have been driven by easier, or more efficient, synthetic access, this remains a somewhat uncomfortable approach and the reader is left to wonder, whether this selection was not also driven by a human-intuition driven decision that the TZDs might perform better than the carboxylic acids.

-Comment on the RORg ligands: The textual clarification has helped appreciating the underlying data.

(Remarks on code availability)

Reviewer #2

(Remarks to the Author)

Peter Willett once told me, "Peer review is far from perfect, but it's the best tool we have so far for promoting quality and novelty in scientific research." With that in mind, I spent a few hours last week searching for literature related to the authors' manuscript. The most relevant studies I found have led me to reconsider my earlier comments about the novelty of this paper. I believe that, to the best of my knowledge, there is no such a study describing the 'incremental transfer learning of generative models for molecule optimisation on potency on a particular target also including experimental validation'.

Overall, the research is novel, and as mentioned in my previous review, the methods are solid. The concept of creating a 'learning trajectory' across several fine-tuning iterations is fascinating although it may not necessarily have greater impact than using scoring methods in practical settings. I would consider this research as cutting-edge which makes this article potentially appropriate for a high-impact journal such as NCOMMS. However, I have now become uncertain on the validity of the experimental data presented in the PPAR design as most of the proposed candidates, claimed to be superior to the most active compound, may contain an assay interferent motif which would invalidate some of the statements of the authors. Note that this does not mean that the method is flawed. I believe the method works but you just presented evidence that a medicinal chemist would likely dismiss in absence of further experimental data. I have included further reply to the authors to improve the quality of the manuscript. On the code side, as I said before, tests are very limited and there is no folder with

scripts to reproduce the results of this manuscript, hence reproducibility is not at its best. This can be a red flag from my side, but the final decision on whether this is a blocker to the publishing of the paper stays with the editor.

- “The previous approaches, though very valuable to advance CLM in drug discovery, have focused mainly on optimizing rather simple parameters such as logP and QED which do not reflect the complexity of drug design.” Note that contributions like that by He et al., *Journal of Cheminformatics* 2022, used transformers for optimising PhysChem-ADME properties, although they lack experimental validation.

- Thanks for including the Supplementary Figures 1 and 2. In particular to Suppl. Fig. 2, I don't know how significant the shift in perplexity (from 1.013 to 1.018) may be for reflecting the ignorance of the PPAR- and NR1-ignorant models, but I will take that there is a trend.

- “[...] spanning more than three orders of magnitude in potency (Fig. 3a, the full fine-tuning set details are shown in Suppl. Tab. 1) with A48 as the most potent entity.”. A is introduced here for the first time but there is no reference to Figure 3e. It would help the reader finding the compound without effort.

- Figure 2g. Not much is being said about the equation in the caption. There is also a format mismatch between NH and Nhold. Moving to the corresponding text and expanded section about the rediscovery score, I still believe that the authors could make a better effort in explaining how the score works to the reader. The sentence “The score considers the number of retrieved known highly active molecules in the 30 most probable designs and their sampling frequency in 2048 generated SMILES strings” is not satisfactory. The reader should understand what the score does.

- Why are you dividing the N of rediscovered molecules by 30?

- Why did you decide to sum the terms rather than multiplying them? Multiplication is used in maths to rapidly decrease scores to zero when any components of an equation are equal or close to zero. In this case, you're only summing terms hence extreme cases can still prevail.

- This being said, I do not mean to say that the score is wrong. I mean that as a reader, I do not get the rationale behind it, hence the authors should try to explain it better since their conclusions rely on their own bespoke score.

- “The stepwise fine-tuning adds an additional dimension in the training data as the model consecutively sees molecules with increasing potency. This matches well the ability of recurrent neural networks to capture sequence-based data.” To my understanding of neural networks, I believe that this statement is wrong. Performing iterative fine tuning (with increasing potency) is not like adding an additional dimension but it may help shaping the optimization trajectory differently – avoiding/reducing gradient shock from sudden domain shifts. By starting with lower-potency molecules and gradually introducing higher-potency ones, the model experiences a smoother gradient landscape. Also, staged fine-tuning should allow each step to reuse learned chemical representations rather than overwriting them abruptly. That's why I believe this may work better than all-in-one go. You are free to take it and do more checks on your side but I would review your statement.

- “While 1 and 7 additionally displayed slightly superior potencies”. A medicinal chemist would specify that “these potencies are within the same order”. And the most problematic bit for me is that you state as shown in Fig 3i, these compounds all have superior PPAR activation at 10 nM (which could be marked in the figure since the scale is M there), which is true, but let's not forget that you have replaced the carboxylic group of compound A with a rhodanine group which is a famous motif responsible for assay interference and flagged as a PAINS as well (<https://pubs.acs.org/doi/full/10.1021/acs.jmedchem.5b00361>). Do you provide evidence that your compounds are actual binders or may they be interfering with your assay at lower concentrations?

- For other comments that I have not addressed specifically in this reply, I want to thank the authors for taking time to account for my feedback and having extended or rephrased some section of the manuscript.

(Remarks on code availability)

I didn't try to install and run the code because there is no point. The code seems to be in place, I can read that it is there and is written decently. However, the repository would benefit from tagging on the version of code used to generate the results of the paper to avoid future variations from invalidating reproducibility. And most importantly, as I said in my comments, there are no scripts to regenerate the results which prevents from reproducing them.

Version 1:

Reviewer comments:

Reviewer #1

(Remarks to the Author)

The authors are congratulated on this nice set of data. The additional evaluation of the carboxylic acids for PPAR has strongly strengthened the storyline and provides extra confidence in the data sets. Combined with the other textual changes, the manuscript, in the view of this reviewer, has become ready for publication.

(Remarks on code availability)

Reviewer #2

(Remarks to the Author)

Thanks for taking time to revise the manuscript. And thanks for pointing out that your compounds motifs are not necessarily PAINS. However, that motif does something to the assay, and you can see that in your response curves. As soon as you replace the thiazolidinedione with the carboxylic acid, the activation jumps 5-10x. It'd be interesting to find out what that groups does. However, this is outside the scope of this paper or my review. I am happy with the changes you made and the data you've integrated. I will support the publication of this manuscript on NCOMMS.

(Remarks on code availability)

Code is not great as I said but the manuscript is solid now.

München, 15.01.2026

Reg.: Revisions for "Structural optimization of drug molecules by incrementally trained language models"

Dear Reviewers

Thank you very much for evaluating our revised manuscript on "Structural optimization of drug molecules by incrementally trained language models" and for your positive feedback. Your input was very valuable to strengthen the manuscript further. We have addressed all your comments as outlined below. Most notably, we have prepared five additional CLM designs from the top-10 list as suggested, and in vitro testing confirmed all five compounds as highly active PPAR γ agonists with sub-nanomolar to low nanomolar potencies. Together with the previously prepared examples, all nine tested designs from the top-10 list were highly active on the target of interest and exceeded the reference A in efficacy (up to ~ 6-fold) and/or in potency (up to ~ 60-fold). We feel that these results impressively corroborate our incremental fine-tuning approach for structural optimization of drug molecules. We hope the revised manuscript meets your expectations and we thank you for your further consideration.

Sincerely

Daniel Merk

REVIEWER COMMENTS

Reviewer #1 (Remarks to the Author):

The manuscript "Structural optimization of drug molecules by generative deep learning" has become significantly clearer because of the changes and additions made. Below, a point-by-point reflection is provided on the initial concerns raised in light of the amended manuscript.

We thank the Reviewer for evaluating our revised manuscript and for the positive feedback. We have addressed the Reviewer's remaining concerns as discussed below.

-Comment on the setting with regards to previous work: The authors have strongly improved the introductory setting of their manuscript, which now allows the reader to appreciate their work much better.

We thank the Reviewer for supporting our revised introduction.

-Comment on the CLM approach: additions to the manuscript have provided more clarity for the choice of LSTM by the authors.

We thank the Reviewer for appreciating the suitability of LSTM for our study.

-Comment on the PPAR γ ligands: The textual clarifications added to the manuscript help in appreciating the approach taken by the authors and in appreciating the data. Notwithstanding, the selection of compounds 1, 3, 7, and 10 for synthesis and testing (out of the ten top-ranking compounds) gives the reader a sense of (strong) bias: these are all TZDs, while none of the carboxylic acids (2, 5, 6, 8, 9), which were similarly ranked in the top ten, were selected for synthesis and testing. Even though the selection of only the TZDs might potentially have been driven by easier, or more efficient, synthetic access, this remains a somewhat uncomfortable approach and the reader is left to wonder, whether this selection was not also driven by a human-intuition driven decision that the TZDs might perform better than the carboxylic acids.

We thank the Reviewer for raising this point. The choice of the TZD derivatives as first LSTM designs to make was indeed partly driven by their easier synthetic accessibility. We agree that this structural modification may contribute to the enhanced potency of the compounds but, in our opinion, this only underscores that the model extracted the relevant structural features for strong activity from the training data. Nevertheless, we have now also prepared and tested the carboxylic acid derivatives on ranks 2, 5, 6, 8 and 9 of the design list which consistently displayed sub-nanomolar to low nanomolar PPAR γ agonism (see Figure 3) and hence exceeded the most active known compound of the series (compound A) by up to a factor of 60 in potency. With the additional experiments performed in this revision, nine out of nine CLM designs prepared from the top-10 list showed either enhanced efficacy or enhanced potency compared to A. In our opinion, these results impressively confirm that the incremental fine-tuning approach performs well in designing highly potent analogues of a given bioactive scaffold.

-Comment on the ROR γ ligands: The textual clarification has helped appreciating the underlying data.

We thank the Reviewer for this remark.

Reviewer #2 (Remarks to the Author):

Peter Willett once told me, “Peer review is far from perfect, but it’s the best tool we have so far for promoting quality and novelty in scientific research.” With that in mind, I spent a few hours last week searching for literature related to the authors’ manuscript. The most relevant studies I found have led me to reconsider my earlier comments about the novelty of this paper. I believe that, to the best of my knowledge, there is no such a study describing the ‘incremental transfer learning of generative models for molecule optimisation on potency on a particular target also including experimental validation’. Overall, the research is novel, and as mentioned in my previous review, the methods are solid. The concept of creating a ‘learning trajectory’ across several fine-tuning iterations is fascinating although it may not necessarily have greater impact than using scoring methods in practical settings. I would consider this research as cutting-edge which makes this article potentially appropriate for a high-impact journal such as NCOMMS . However, I have now become uncertain on the validity of the experimental data presented in the PPAR design as most of the proposed candidates, claimed to be superior to the most active compound, may contain an assay interferent motif which would invalidate some of the statements of the authors. Note that this does not mean that the method is flawed. I believe the method works but you just presented evidence that a medicinal chemist would likely dismiss in absence of further experimental data. I have included further reply to the authors to improve the quality of the manuscript. On the code side, as I said before, tests are very limited and there is no folder with scripts to reproduce the results of this manuscript, hence reproducibility is not at its best. This can be a red flag from my side, but the final decision on whether this is a blocker to the publishing of the paper stays with the editor.

We thank the Reviewer for this encouraging and honest feedback. We are very grateful for the Reviewer’s critical evaluation of our work. We have addressed the additional comments as outlined below.

- “The previous approaches, though very valuable to advance CLM in drug discovery, have focused mainly on optimizing rather simple parameters such as logP and QED which do not reflect the complexity of drug design.” Note that contributions like that by He et al., Journal of Cheminformatics 2022, used transformers for optimizing PhysChem-ADME properties, although they lack experimental validation.

We thank the Reviewer for spotting this error. We have revised the introduction accordingly.

- Thanks for including the Supplementary Figures 1 and 2. In particular to Suppl. Fig. 2, I don’t know how significant the shift in perplexity (from 1.013 to 1.018) may be for reflecting the ignorance of the PPAR- and NR1-ignorant models, but I will take that there is a trend.

We thank the Reviewer for this comment. Supplementary Figure 2 has been included to show that PPAR- and NR1-ignorant models are not significantly different with respect to their perplexity on known PPAR ligands which underscores that removing PPAR ligands from the pre-training is sufficient to remove any pre-bias for the target in the model. Compared to the model trained with all ChEMBL data (including PPAR/NR1), both the PPAR- and NR1-ignorant model display significantly ($p < 0.001$, ANOVA) higher perplexity. The difference in perplexity between the PPAR-ignorant and the NR1-ignorant model is negligible (though statistically significant due to large sample size). We have included these statistical results in the figure.

- “[...] spanning more than three orders of magnitude in potency (Fig. 3a, the full fine-tuning set details are shown in Suppl. Tab. 1) with A48 as the most potent entity.”. A is introduced here for the first time but there is no reference to Figure 3e. It would help the reader finding the compound without effort.

We thank the Reviewer for this remark. We have added reference to the corresponding figure to help the reader find compound A.

- Figure 2g. Not much is being said about the equation in the caption. There is also a format mismatch between NH and Nhold. Moving to the corresponding text and expanded section about the rediscovery score, I still believe that the authors could make a better effort in explaining how the score works to the reader. The sentence “The score considers the number of retrieved known highly active molecules in the 30 most probable designs and their sampling frequency in 2048 generated SMILES strings” is not satisfactory. The reader should understand what the score does.

We thank the Reviewer for the further critical comments on the rediscovery score. We have addressed them as follows.

We have updated Figure 2g to correct the format mismatch and further revised the text explaining the score to be more clear.

- Why are you dividing the N of rediscovered molecules by 30?

The number of rediscovered molecules is divided by 30 as the top 30 LSTM designs based on perplexity are analyzed. This factor of the score thus considers the fraction of rediscovered molecules in the top-ranking designs of the analyzed model. We have revised the text to clarify this.

- Why did you decide to sum the terms rather than multiplying them? Multiplication is used in maths to rapidly decrease scores to zero when any components of an equation are equal or close to zero. In this case, you’re only summing terms hence extreme cases can still prevail.

The Reviewer is perfectly right, that multiplying the terms instead of summing them up (i.e., using the geometric mean instead of the arithmetic mean) would also be reasonable. We have now compared both approaches. Changing the score to the product of the three terms did not notably change the results of our study, indicating that both approaches work. In line with the Reviewer’s suggestion, we have revised the manuscript to use the score as the geometric mean of the three terms, as this may indeed better avoid overweighting in other scenarios/applications.

- This being said, I do not mean to say that the score is wrong. I mean that as a reader, I do not get the rationale behind it, hence the authors should try to explain it better since their conclusions rely on their own bespoke score.

We thank the Reviewer again for critically evaluating our scoring approach and for the valuable feedback that has helped improving the score and its presentation/explanation.

- “The stepwise fine-tuning adds an additional dimension in the training data as the model consecutively sees molecules with increasing potency. This matches well the ability of recurrent neural networks to capture sequence-based data.” To my understanding of neural networks, I believe that this statement is wrong. Performing iterative fine tuning (with increasing potency) is

not like adding an additional dimension but it may help shaping the optimization trajectory differently – avoiding/reducing gradient shock from sudden domain shifts. By starting with lower-potency molecules and gradually introducing higher-potency ones, the model experiences a smoother gradient landscape. Also, staged fine-tuning should allow each step to reuse learned chemical representations rather than overwriting them abruptly. That’s why I believe this may work better than all-in-one go. You are free to take it and do more checks on your side but I would review your statement.

We thank the Reviewer very much for sharing this perspective. We agree that this another very convincing interpretation of the improved performance. We have revised the manuscript (Results and Discussion sections) to consider and discuss several possible explanations including the proposed improved learning/optimization trajectory with smoother gradient.

- “While 1 and 7 additionally displayed slightly superior potencies”. A medicinal chemist would specify that “these potencies are within the same order”. And the most problematic bit for me is that you state as shown in Fig 3i, these compounds all have superior PPAR activation at 10 nM (which could be marked in the figure since the scale is M there), which is true, but let’s not forget that you have replaced the carboxylic group of compound A with a rhodanine group which is a famous motif responsible for assay interference and flagged as a PAINS as well (<https://pubs.acs.org/doi/full/10.1021/acs.jmedchem.5b00361>). Do you provide evidence that your compounds are actual binders or may they be interfering with your assay at lower concentrations?

We thank the Reviewer for this important comment but respectfully disagree that the observed activity differences are not meaningful and that the selected CLM-designed compounds would be flagged by medicinal chemists as PAINS:

- While the potencies of the most active template A and the CLM designs are in a similar range, they do not reflect efficacy which is substantially higher for the LSTM-designs.

- Regarding PAINS: The CLM-designed compounds contain a thiazolidinedione motif which is related to rhodanine, but the PAINS issue of compounds with rhodanine related substructures arises from an exocyclic double bond (shown on the right below; see 10.1021/jm901137j and 10.1021/acs.jmedchem.5b00361). Such motif is not present in the compounds prepared and tested in our study, and a PAINS character is hence not given.

Thiazolidinedione
(compounds in our study)

Rhodanine

exocyclic double bond
potential PAINS

Nevertheless, we have now prepared the carboxylic acid derivatives from ranks 2, 5, 6, 8 and 9 of the CLM design list to extend the prospective results. All five designs were highly active and exhibited sub-nanomolar to low nanomolar potency on PPAR γ thus exceeding the most active known compound A in potency by 12- to 60-fold (which are meaningful differences). Together with the previous compounds, these results demonstrate that the CLM can design highly active analogues (not PAINS) and impressively corroborate the performance of the incremental training.

- For other comments that I have not addressed specifically in this reply, I want to thank the authors for taking time to account for my feedback and having extended or rephrased some section of the manuscript.

We thank the Reviewer for this positive feedback and for the valuable input that has inspired these revisions.

Remarks on code availability:

I didn't try to install and run the code because there is no point. The code seems to be in place, I can read that it is there and is written decently. However, the repository would benefit from tagging on the version of code used to generate the results of the paper to avoid future variations from invalidating reproducibility. And most importantly, as I said in my comments, there are no scripts to regenerate the results which prevents from reproducing them.

We thank the Reviewer for this comment. We ensure that the code and data are provided to reproduce and extend our work.